# *Plasmodium falciparum* Subtilisin-like Domain-Containing Protein (PfSDP), a Cross-Stage Antigen, Elicits Short-Lived Antibody Response Following Natural Infection with *Plasmodium falciparum*

**DOI:** 10.3390/cells14151184

**Published:** 2025-07-31

**Authors:** Jonas A. Kengne-Ouafo, Collins M. Morang’a, Nancy K. Nyakoe, Daniel Dosoo, Richmond Tackie, Joe K. Mutungi, Saikou Y. Bah, Lucas N. Amenga-Etego, Britta Urban, Gordon A. Awandare, Bismarck Dinko, Yaw Aniweh

**Affiliations:** 1West African Centre for Cell Biology of Infectious Pathogens, College of Basic and Applied Sciences, University of Ghana, Legon, Accra P.O. Box LG 56, Ghana; jakengne@gmail.com (J.A.K.-O.); collinsmoranga@ug.edu.gh (C.M.M.); kemuma2007@gmail.com (N.K.N.); dosoodaniel@gmail.com (D.D.); joemutungi@gmail.com (J.K.M.); saikouybah@gmail.com (S.Y.B.); lamengaetego@ug.edu.gh (L.N.A.-E.); gawandare@ug.edu.gh (G.A.A.); 2Department of Clinical Microbiology, School of Medicine and Dentistry College of Health Sciences, Kwame Nkrumah University of Science and Technology, Kumasi 00233, Ashanti Region, Ghana; rtackie@uhas.edu.gh; 3Department of Tropical Disease Biology, Liverpool School of Tropical Medicine, Liverpool L3 5QA, UK; britta.urban@lstmed.ac.uk

**Keywords:** *Plasmodium falciparum*, clinical isolates, antibody response, genetic diversity, gene expression, localization

## Abstract

With the increasing detection of artemisinin resistance to front-line antimalarials in Africa and notwithstanding the planned roll-out of RTS’S and R21 in Africa, the search for new vaccines with high efficacy remains an imperative. Towards this endeavour, we performed in silico screening to identify *Plasmodium falciparum* gametocyte stage genes that could be targets of protection or diagnosis. Through the analysis we identified a gene, Pf3D7_1105800, coding for a *Plasmodium falciparum* subtilisin-like domain-containing protein (PfSDP) and thus dubbed the gene *Pfsdp*. Genetic diversity assessment revealed the *Pfsdp* gene to be relatively conserved across continents with signs of directional selection. Using RT qPCR and Western blots, we observed that *Pfsdp* is expressed in all developmental stages of the parasite both at the transcript and protein level. Immunofluorescence assays found PfSDP protein co-localizing with PfMSP-1 and partially with Pfs48/45 at the asexual and sexual stages, respectively. Further, we demonstrated that anti-PfSDP peptide-specific antibodies inhibited erythrocyte invasion by 20–60% in a dose-dependent manner, suggesting that PfSDP protein might play a role in merozoite invasion. We also discovered that PfSDP protein is immunogenic in children from different endemic areas with antibody levels increasing from acute infection to day 7 post-treatment, followed by a gradual decay. The limited effect of antibodies on erythrocyte invasion could imply that it might be more involved in other processes in the development of the parasite.

## 1. Introduction

*Plasmodium falciparum* is increasingly becoming resistant to frontline antimalarials [1,2] whilst no effective vaccine exists against malaria. Even though the circumsporozoite protein-based vaccines (RTS’S-Mosquirix, R21/Matrix-M) have been approved, their efficacy remains low (https://www.malariavaccine.org/existing-vaccines/rtss (accessed on 15 June 2025)). RTS, S/AS01 and R21/Matrix-M target the parasite’s sporozoite stage and are recommended for children in endemic regions [3]. RTS, S/AS01 showed approximately 55% efficacy against clinical malaria and 30% against severe malaria in trials, with real-world pilots demonstrating a 13% reduction in all-cause child mortality [4]. R21/Matrix-M exhibited higher efficacy, reaching 78% against uncomplicated malaria over 12 months in some trials and provided significant protection when combined with seasonal chemoprevention [5,6]. The protection provided by both vaccines wanes over time unless a booster dose is administered. Their integration with existing malaria control measures is crucial as research continues on more advanced vaccine candidates [7]. The challenge in malaria vaccine development and its efficacy is largely due to the complexity of the parasite’s life cycle, genetic diversity, and antigenic variations [8]. In the human host, the parasite invades liver cells and red blood cells. Additionally, infected cells sequester in tissues with late asexual stages found mostly in the spleen and gametocytes in the bone marrow [9,10,11,12]. The asexual stages are exclusively found in the human host, whereas the sexual stages are initiated in the human host [9,10,13] but continue their development in the mosquito following a gametocyte-infected blood meal [14,15]. Tackling both the asexual and sexual stages of the parasite could be an effective strategy to control malaria. One way to attain such a goal is the development of multi-component vaccines having both anti-disease and transmission-blocking activities [16,17]. Another way to achieve this will be to select antigens from different stages of the parasite’s life cycle or antigens with cross-stage expression. This has become a hurdle due to the polymorphic nature of some of the antigens [18]. With the advent of the X-omics technology, parasite-related genomic, transcriptomic, and proteomic data have increased exponentially over the past years [19]. In silico analysis of such data shows that a majority of genes remain uncharacterized, with many lacking classical motifs that would suggest their function [20]. Characterizing these genes could help identify functionally relevant ones that can serve as vaccine or drug targets [21]. In addition to its immunogenicity, a good vaccine target should be conserved to offer strain- or/and species-transcending protection and accessible to the immune system to induce appropriate responses [22,23]. Nowadays, reverse vaccinology approaches have been developed to predict the subcellular localization of proteins and their potential as vaccine candidates [22,24]. The present study aimed to utilize available *P. falciparum* genomics, transcriptomics, and proteomics data to discover and prioritize genes with potential vaccine candidate properties. The selected candidates were assessed for their genetic diversity across continents and their role in the biology of the parasite. Also, the localization, breadth, and kinetics of the antibody response of PfSDP were assessed. This study demonstrated that the *PfSDP* gene is conserved across continents and encodes for a subtilisin-like domain-containing protein. It co-localizes with *P. falciparum* merozoite surface protein-1 (PfMSP-1) in asexual stage schizonts. PfSDP protein is immunogenic in children with antibodies decaying from 2 weeks post-infection. Antibodies to PfSDP-specific peptides inhibited red blood cell invasion by merozoites.

## 2. Materials and Methods

### 2.1. Gene Selection

Genes were prioritized from available *P. falciparum* proteome and gene expression datasets (unpublished, either provided by Dr Yaw Aniweh or found in PlasmoDB). The primary screen was achieved by filtering all differentially expressed genes with unknown function. This was followed by sorting out the genes predicted to have signal peptides and/or transmembrane domains using the protein features on the PlasmoDB (http://plasmodb.org/plasmo/app/record/gene (accessed on 15 June 2019)) platform. The different expression patterns of the genes across erythrocytic stages and their essentiality were also checked on PlasmoDB.

### 2.2. Genetic Diversity Assessment

*Plasmodium falciparum* genomes from the Pf3k MalariaGEN database release 3 (https://www.malariagen.net/data_package/pf3k-pilot-data-release-3/ (accessed on 6 June 2025)) were used to assess the genetic diversity of selected genes Genome data of *P. falciparum* samples collected from 7 different countries in Africa (Congo, Nigeria, Malawi, Mali, Ghana, Senegal, and Guinea) and 5 in South-East Asia (SEA) (Laos, Vietnam, Bangladesh, Cambodia, and Thailand) were used. Bam files were indexed, and the genomic regions covering the selected genes were extracted as reads using samtools (v1.4) [25], while the genomic location of the gene of interest was obtained from the reference genomic feature file (GFF). The reads were used to generate the consensus sequences using a custom script involving samtools (v1.4) and bcftools (v.1.41). The consensus sequences were subsequently processed using seqtk tool (v1.2), aligned using MAFFT (v7.4) (Fast Fourier Transform-auto (mafft_auto) algorithm [26], and converted to phylip format using a custom-made script. RAxML (v0.1) software was used to generate phylogenetic trees for the gene. The best phylogenetic tree was used to determine cophenetic distances, which are a measure of similarity or closeness between sequences. A heatmap was then plotted using cophenetic correlation coefficients generated by computing the correlation levels in the distance matrix. Nucleotide variations between continents were measured using the Fst test. Neutrality tests (Tajima’s D) were used to check whether the selected genes were under selection. These were conducted using the PopGenome package (v2.7.1) [27]. Haplotype network analysis was carried out using the Pegas package (v1.2). The PopGenome and Pegas were implemented in the open-source R-programming software version 3.4 (https://www.r-project.org/ (accessed on 15 June 2020)).

### 2.3. Description and Source of Clinical Isolates and Lab Strains

Clinical *P. falciparum* isolates and the reference strains, NF54 and 3D7, were used in this study [28]. Clinical isolates were sampled from symptomatic children aged 2 to 14 years, who reported at the Ledzokuku-Krowor Municipal Assembly Hospital (LEKMAH) in Teshie, Accra (Greater Accra Region), Ghana. The samples were collected in a major study with ethical approval numbers GHS-ERC:002/08/17 and GHS-ERC:03/09/16. Before sample collection, the objective, nature, and potential risks associated with this study were well explained to the participants and/or guardians who gave informed consent.

#### Parasite Culture, Developmental Stage Production

Frozen parasitized RBCs were thawed using 2 different concentrations (12% and 1.6% consecutively) of NaCl following a protocol previously described by [28]. The thawed parasites were then washed 3 times in RPMI 1640 medium (Sigma, Gillingham, UK) to remove lysed cells and any other unwanted components. Washed parasitized cells were maintained in culture at 2% hematocrit in complete culture medium (RPMI 1640 medium (Sigma, Gillingham, UK), supplemented with 0.5% Albumax II (Gibco, London, UK), 50 ug/mL of Gentamicin and 2 mg/mL of sodium bicarbonate) containing 2% heat-inactivated normal human serum (NHS) (PAN Biotech, Wimborne, UK). Before incubation at 37 °C, the culture flasks (CORNING) were gassed with a 5.5% CO_2_, 2% O_2_, and N_2_ gas mixture. The cultures were maintained in O^+^ RBCs. Gametocyte production was carried out as described by Fivelman et al. [29]. Prior to gametocytogenesis induction, parasites were synchronized using 5% D-Sorbitol (Sigma Aldrich, Gillingham, UK) to ring stage parasites [29]. Gametocytogenesis was induced by increasing the hematocrit and feeding the parasites with a mixture of fresh and spent media in a ratio of 2:3 on the first day of induction (Day-2), 1:2 on the second day (Day-1), and with fresh media on the third day (Day-0). From day 4 the cultures were maintained in complete medium (CM) supplemented with 50 mM of N-acetyl glucosamine (SIGMA, Gillingham, UK). Parasitaemia was monitored daily by thin smear Giemsa staining and microscopy. At days 6 and 12, early and late-stage gametocytes were pelleted by using Percoll gradient centrifugation (3 mL of 90%, 2 mL of 70%, and 40% layered from bottom to top) at 2400 rpm for 20 min. Late-stage (stages IV and V) gametocytes were collected from the 40%/70% Percoll gradient. The collected samples were washed twice in RPMI medium at 1500 rpm for 10 min and stored in Trizol at −80 °C until RNA extraction.

### 2.4. RNA Extraction, RT q-PCR

RNA extraction was performed using Direct-zol RNA miniprep Kits (ZYMO RESEARCH) following the manufacturer’s instructions, and concentration was checked using a NanoDrop spectrophotometre (Thermo Fisher Scientific, Madison, WI, USA). The expression profiles of the selected genes in *P. falciparum* gametocytes and asexual stages were analyzed using the Luna^®^ Universal One-Step RT-qPCR Kit (New England Biolabs, Inc., Ipswich, MA, USA). The reactions were performed in triplicates in a final volume of 10 µL following the manufacturer’s instructions. Experiments were performed on a QuantStudio 5 Real-Time PCR System (Applied Biosystems, Waltham, MA, USA). All reactions were run alongside a reference gene (*P. falciparum* seryl-tRNA synthetase). The fold change in gene expression in gametocyte and asexual stages was estimated using the formula 2^−ΔΔCT^ [30] and asexual stages (rings) as reference. Primer design for PF3D7_1105800 was carried out using A plasmid Editor (ApE) software (v2.0.30) (https://jorgensen.biology.utah.edu/wayned/ape/ (accessed on 20 May 2025)) with the cDNA sequence downloaded from PlasmoDB (forward 5′-GCATTAAGTTTAGCAGGTGGTG-3′ reverse 5′-GCTTCCACATCTTCTGACGT-3′). Primer sequences for *P. falciparum* seryl-tRNA synthetase and Pfs25 were used as positive control [31,32]. A melt curve was performed on the final product to determine the specificity of the primers (Appendix A).

### 2.5. Antibody Production

The 3D7 strain amino acid sequence of the PF3D7_1105800 protein was downloaded from PlasmoDB and submitted to the GenScript platform for B cell epitope mapping and synthesis (GenScript, Piscataway, NJ, USA). Three peptides were selected based on the predicted sequence antigenicity, surface exposure, and hydrophobicity scores from the GenScript OptimumAntigenTM design tool. Peptide 1 corresponded to amino acid residues 31–45 from the protein’s N-terminal region, while peptide 2 comprised residues 160–174, and peptide 3 consisted of residues 94–108. These peptides were used to immunize New Zealand Rabbits by GenScript to produce polyclonal antibodies. These antibodies were purified by passing immune sera from the animals through a column containing A/G coupled protein beads followed by concentration through an Amicon 30 kDa and reconstituted in PBS (GenScript, Piscataway, NJ, USA).

### 2.6. Immunoblotting

Immunoblotting was carried out to confirm the gene expression at the protein level. To achieve that, ring, trophozoite, schizont, and early and mature gametocyte stage pellets were produced as previously described. To eliminate hemoglobin, the pellets were incubated in 5 mL of cold 0.03% saponin (SIGMA, Gillingham, UK) solution for 10 min at RT on ice and washed repeatedly with 1X PBS till the supernatant became clear [33]. The pellets were subsequently lysed in 1% triton_X-100 lysis buffer (1% triton_X-100, 500 mM of tris-HCl, pH 7.5, 150 mM of NaCl, 1 mM EDTA, and 1X Halt™ Protease Inhibitor Cocktail (Thermo Scientific, Cambridge, UK)) [33]. Uninfected RBCs were subjected to the same procedure to produce ghost membranes that served as negative control. The 6X SDS loading buffer dye was added to 50 µL of each sample, heated at 95 °C for 15 min. In total, 20 µL was loaded alongside molecular markers on a 12% pre-cast SDS-PAGE gel (GenScript, Oxford, UK) and run at 100 V. Proteins were then blotted onto a nitrocellulose membrane (BIO-RAD, Feldkirchen, Germany) for 20 min using a trans-blot Turbo transfer system (BIO-RAD, Feldkirchen, Germany) at 20 V following manufacturer’s instructions. After transfer, the membrane was washed 3 times in 1X tris-buffered saline (TBS) and 0.1% Tween 20 (TBST) and then blocked in 3% bovine serum albumin (BSA) overnight. The membrane was then immuno-stained with primary antibodies (raised against peptides specific to our proteins of interest (GenScript, Oxford, UK)) at a dilution of 1:500. After washing steps, the membrane was stained in mouse anti-rabbit HRP conjugated secondary antibody (Thermo Scientific, Cambridge, UK) at 1:10,000 dilution. The SuperSignal West Pico Plus chemiluminescent substrate (Thermo Scientific, Cambridge, UK) was used for signal development, and images were captured in a chemiluminescent Amersham Imager 600 (AI600) (GE Healthcare, Little Chalfont, UK).

### 2.7. Full-Length Protein Expression and Purification

Complementary DNA (cDNA) coding for the PF3D7_1105800 proteins was codon-optimized, synthesized, and subcloned by GenScript (Piscataway, NJ, USA) in frame with a C-terminal Hexa-histidine (6x His) tag into a T7 promoter *Escherichia coli* expression vector (pET-24b) to obtain a chimera plasmid with improved expression in *E. coli*. BL21-Codon Plus (DE3)-RIPL *E. coli* competent cells (Agilent Technologies, Santa Clara, CA, USA) were transformed with the chimera plasmids and cultured in terrific broth (12 g of bacto-tryptone, 24 g of bacto-yeast extract, and 4 mL of glycerol per litre) supplemented with 50 µM of kanamycin till an optical density of 0.5 was obtained. Protein expression was induced by the addition of 0.25 mM of Isopropyl β-D-1-thiogalactopyranoside (IPTG) into the culture and shaking at RT for 16 h. Crude lysate was extracted using a phosphate-based lysis buffer (50 mM of NaH2PO4, 300 mM of NaCl, 10 mM of imidazole, pH 8.0). To purify the target proteins, the lysates were passed through Ni-NTA resin-containing column (Qiagen, Germantown, MD, USA) and eluted in a gradient of imidazole [34]. The imidazole buffer was then exchanged with 1X PBS using 10 kDa Amicon centrifugal filters (Merck, Rahway, NJ, USA). The recombinant proteins were subsequently subjected to size exclusion chromatography (GE, Superdex-200 increase 10/300 GL column) using the AKTA PURE (Cytiva, Uppsala, Sweden). The samples were subjected to SDS-PAGE using a 12% pre-cast SDS-PAGE gel (GenScript, Oxford, UK) and stained with Coomassie Brilliant Blue dye. Bands from the gels were cut, processed, and subjected to Liquid Chromatography–Mass Spectrometry (LC-MS) to validate the identity of the protein.

### 2.8. Protein Structure Prediction

The 3D7 strain amino acid sequences of PfSDP were submitted to the I-TASSER platform for 3-D structure prediction. Data generated by I-TASSER was visualized and labelled using PyMOL package (v2.3). Moreover, using I-TASSER output data and protein structural information obtained from the Eukaryotic Linear Motif resource for Functional Sites in Proteins (ELM), the topographical representation of the proteins was designed.

### 2.9. Indirect Immunofluorescence Assay (IFA)

Thin smears of *P. falciparum* asexual and sexual stages were prepared, air-dried, and then fixed with cooled 4% paraformaldehyde-PBS at RT for 10 min and then kept at −80 °C till use. Before IFA, smears were acclimatized to RT for 10 min, permeabilized with 0.1% Triton X-100 dissolved in 1X PBS, and then blocked for 1 h with 1% BSA-PBS for 30 min. The smears were co-stained with the target protein anti-peptide primary antibodies at 1:500 dilution and antibodies for the organelle markers (anti-merozoite surface protein-1 (MSP-1); anti-Pfs48/45 (BEI Resources MR4, Manassas, VA, USA) at 1:1000 dilution was added and incubated for 2 h at RT in a humidified chamber. After 3 washes of slides in PBS-0.1%Triton, goat anti-rabbit Alexa Fluor^®^ 488 and goat anti-mouse Alexa Fluor^®^ 568 (Invitrogen, Paisley, UK) secondary antibodies, at a dilution of 1:500, were added and incubated for 1 h as previously described. The slides were washed and mounted with VECTASHIELD antifade mounting medium (Burlingame, CA, USA) containing 4′, 6′-diamidino-2-phenylindole (DAPI) for nucleus staining and sealed with coverslips and nail polish. Differential interference contrast (DIC) and fluorescent images were captured using Carl Zeiss LSM800 confocal microscope (Carl Zeiss Microscopy GmbH, Jena, Germany). The obtained images were processed using the open-access Fiji-Image J software v1.53t). 

### 2.10. Growth Inhibition Assays (GIAs)

In vitro growth inhibition assays were performed using antibodies (GenScript, Piscataway, NJ, USA) raised against PfSDP-specific peptides following a previously published protocol [20]. Briefly, late trophozoite- or schizont-infected RBCs were mixed with an equal volume of carboxyfluorescein diacetate succinimidyl ester (CFDA-SE)-stained (20 µM) O^+^ RBCs and cultured in complete RPMI supplemented with increasing concentrations (0–250 mg/mL) of each antibody. Cultures were maintained for about 24–30 h at 37 °C in 96-flat-bottom well plates. The experiments were carried out in 50 µL in duplicates. In addition to the target antibodies, anti-human basigin antibodies (clone MEM-M6/6, Thermo Scientific, Cambridge, UK) and pre-immune serum sample (GenScript, USA) were used as positive and negative controls, respectively. Anti-basigin antibodies were used at a working concentration of 12 µg/mL [35]. After incubation, plates were centrifuged at 1500 rpm for 5 min, and the supernatants were carefully removed and replaced with 45 µL of 1 µg/mL Hoechst 33342 (Thermo Scientific, Cambridge, UK) solution for parasite DNA staining. The data was then acquired using a BD LSR Fortessa X-20 cytometer (BD Biosciences, Erembodegem, Belgium), and the invasion rates were determined.

### 2.11. PfSDP Immunogenicity Assessment by Indirect-ELISA

To assess the immunogenicity and kinetics of PF3D7_1105800-specific antibodies, archived plasma samples collected from areas with different transmission intensities in Ghana (Navrongo, Kintampo, Accra, and Ho) were used. These samples were collected from symptomatic and asymptomatic individuals in the framework of projects based at the West African Centre for Cell Biology of Infectious Pathogens (WACCBIP). Some of the projects (Kintampo, Accra, and Ho) were longitudinal. In Ho, follow-up extended only to day 7, while the project in Navrongo involved a cohort study design in which samples were collected from adults and asymptomatic (sub-clinical infection) individuals. Apart from these minor differences in the study design, sample collection was performed following the same protocol.

Using indirect ELISA, protein-specific IgGs were measured in plasma samples. Before the proper assay, checkerboard titration was used to determine the optimum sample and conjugate antibody dilution. Briefly, the 96-well flat-bottom plates (NUNC, Roskilde, Denmark) were coated with 1 μg/mL of PF3D_1105800 recombinant proteins and incubated overnight at 4 °C. The plates were washed 3 times with 1× PBS supplemented with 0.05% Tween-20 (PBS-T) and blocked with 200 µL/well of 5% bovine serum albumin (BSA, SIGMA, St Louis, MO, USA) in PBS-T for 1 h at RT. The plates were washed once more as previously described, and 100 µL/well of the plasma samples were added at 1:1000 dilution in 1% BSA-PBS-T. The plates were incubated at RT for 2 h followed by washing. Then, 100 µL/well of anti-human HRP-conjugated secondary antibody (Thermo Scientific, Cambridge, UK) was added at 1:8000 in 1% BSA-PBS-T and incubated for 1 h at RT. After the final wash, HRP activity was measured with one-Step™ Ultra 3,3′,5,5′-Tetramethylbenzidine (SIGMA, St Louis, MO, USA) by measuring the optical density (OD) at 450 nm using a Varioskan™ LUX multimode microplate reader (Thermo Scientific, Waltham, MA, USA). For these assays, a non-immune serum negative control (from a malaria-naïve European donor) was used to determine the seroreactivity in the study population. A pooled positive plasma sample was used throughout the assays as standard to estimate antibody levels in the study population and account for inter-plate variations. The standard was used to generate a standard curve from which the IgG level of individual samples was extrapolated from their ODs and converted into arbitrary units (AUs) using the four-parameter logistic curve fitting program known as ADAMSEL (Edmond J. Remarque^®^). The immunogenicity of the expressed proteins was validated using Western and dot blots.

#### Protein Immunogenicity Confirmation by Western and Dot Blots

To confirm immunogenicity of the protein, recombinant proteins were subjected to SDS-PAGE, transferred to nitrocellulose (NC) membrane, and stained with plasma (1:1000) from malaria-exposed and non-exposed individuals followed by goat anti-human HRP-conjugated secondary anti-IgG at a dilution of 1:15,000 (Thermo Scientific, Cambridge, UK) [20]. For Dot blot, 10 µL of different concentrations (5 µg/mL, 10 µg/mL, 15 µg/mL) of the protein was spotted directly on NC membrane and immunostained as previously described. This was carried out to ascertain that plasma from malaria-exposed individuals could detect the recombinant proteins [20].

### 2.12. Data Analysis

The PopGenome (v2.7.1) and Pegas (v1.2) packages were used to assess the genetic diversity of the prioritized genes. Fold changes in gene expression arising from qPCR analysis were compared between parasite developmental stages using Kruskal–Wallis followed by a Dunnett’s post hoc test.

The generated IgG levels were log-transformed and compared between groups (areas with different transmission levels, different age groups) using the Mann–Whitney U and Kruskal–Wallis tests for pairwise and multiple group comparisons, respectively. Dunnett’s multiple comparison test was used for pairwise comparisons following a significant Kruskal–Wallis test. Spearman rank correlation analysis was carried out to determine associations between antibody responses and parasitaemia. The chi-square test was used for categorical variables. All statistical analyses were performed using the R-programming software (https://www.r-project.org/ (accessed on 15 June 2020)) and SPSS 20 (Software SPSS INC., Chicago, IL, USA). Visualization was also achieved using GraphPad Prism v.10. The significance level was set at 5% for all statistical tests.

## 3. Results

### 3.1. Expression Profiles of Selected Genes in Online Datasets (PlasmoDB)

In identifying genes that are expressed across the sexual and asexual stages, we identified *PfSDP* as a possible candidate for validation. The gene is found on chromosome 11. It is conserved with unknown functions and expressed across parasite stages from late trophozoites to schizonts and early to late gametocytes [36,37]. It has also been demonstrated to be involved in protein–protein interactions in *P. falciparum* [38]. In addition, using genome-wide CRISPR screening, the orthologue of this gene in *Toxoplasma gondi* has been found to be essential and likely to be involved in cell invasion [39].

### 3.2. Pfsdp May Be Under Directional Selection and Genetically Conserved Across Different Continents

To query the levels of genetic diversity of the gene, a total of 2276 samples (1336 from Africa and 1010 from South-East Asia, SEA) were selected from the Pf3k MalariaGEN database release 3 (https://www.malariagen.net/data_package/pf3k-pilot-data-release-3/ (accessed on 15 June 2025)). Our target gene of interest, PF3D7_1105800, *PfSDP*, had limited levels of polymorphisms with 91 and 18 segregating sites in Africa and SEA, respectively. Nucleotide diversity (Pi) was quite low both in Africa (0.00168) and SEA (0.00085). The same trend was observed with haplotype diversity (Hd). Generally, the haplotype number was higher in Africa than in SEA, estimated at 101 and 40, respectively). The overall haplotype diversity was higher in Africa (0.916) than SEA (0.873). However, it is worth mentioning that haplotype construction would have been more prone to error in Africa as compared to SEA due to higher multi-genomic infections (multiclonal infections).

Population structure assessment using Fst revealed moderate genetic differentiation: Fst in the range of 0.005–0.15. The gene was found to have negative Tajima’s D values, suggesting directional selection or recent population expansion. This departure from neutrality was more pronounced in Africa than in SEA (Figure 1). The diversity is likely to be more ancient than would be expected under neutrality [40].

To check whether there was any clustering of the observed limited genetic diversity per region, heatmap and principal component analysis (PCA) scatter plots were generated. The heatmap showed that the *pfsdp* diversity was neither region- nor country-specific but rather randomly distributed across countries and continents (Figure 1A). The PCA summarizing this genetic diversity confirmed the conserved nature of this gene as all samples clustered together (Figure 1B). Moreover, haplotype network analysis showed the major haplotypes present in both Africa and SEA (Appendix A), confirming our results.

### 3.3. PfSDP Has Cross-Stage Expression at the Transcript and Protein Levels in Clinical Isolates

To evaluate and compare the expression pattern of Pf3D7_115800 (*Pfsdp*) gene in clinical isolates to that of the 3D7 strain found in PlasmoDB, a total of five isolates were used, four of which were clinical isolates and the NF54 lab reference strain. A total of 29 samples, composed of 5 rings, 5 trophozoites, 5 schizons, 5 early gametocytes, and 9 late gametocytes (5 from clinical isolates and 4 from the NF54 reference strain), were produced. Using total mRNA, PF3D7_115800 was found to have cross-stage expression by quantitative PCR (Figure 2A). As expected, this gene was highly expressed in early gametocytes derived from clinical isolates (stage 2 and 3) with a mean fold change of 8 (Figure 2B). This expression pattern was consistent with that of the laboratory strain reported in PlasmoDB. Moreover, the expression in clinical isolate-derived late gametocytes was like that of their NF54 counterparts used as a control (Figure 2C,D). The positive control, *Pfs25*, had the expected expression pattern of slight expression in asexual stages and early-stage gametocytes and high expression in late gametocytes (500-fold increase in clinical isolate gametocytes compared to their ring counterparts), ascertaining sample purity (Figure 2).

Protein structure prediction reveals PF3D7_1105800 to be made up, mainly, of alpha helices consisting mostly of a transmembrane domain, a disordered region, a low complexity region, a globular domain, and a subtilisin/kexin isozyme-1 (SKI1) domain (Figure 3A,B). This structure is predicted from that of a Photosynthetic Reaction Centre Protein. In addition, orthologues of this gene in *Toxoplasma gondi* are shown, through a genome-wide CRISPR screen, to be essential and likely to be involved in cell invasion [39]. Based on these observations, this protein is referred to as a *P. falciparum* subtilisin-like domain-containing protein (*PfSDP*).

Moreover, Western blots were used to confirm protein expression in different developmental stages of clinical isolates (Figure 3C–E). PF3D7_115800 (*PfSDP*) protein was found to be expressed in all developmental stages and more likely to be dimeric in nature or to undergo processing in the parasite as demonstrated with *PfSDP*-specific anti-peptide 1, 2, and 3 antibodies (Figure 3C–E). Anti-peptide-2 recognized a band (about 37 kD) in all stages with a pronounced signal in early gametocytes. However, this band was not detected in the recombinant protein, suggesting protein processing or interaction in the parasite. This antibody also detected a dimer in late gametocytes. The list and characteristics of selected peptides used for antibody production are shown in Appendix A.

### 3.4. PfSDP Co-Localized with MSP-1, Partially with Pfs45/48, and Peptide-Specific Antibodies Showed Mild Impacts on Merozoite Invasion in a Concentration-Dependent Manner

To determine the localization of the selected protein, various parasite developmental stages were immuno-stained with antibodies, each specific to a region (peptide) on the *PfSDP* protein. Parasites were co-stained with antibodies to known organelle markers: merozoite surface protein 1 (MSP-1) for merozoite (parasite) surface [41,42] and Pfs48/45 for gametocyte surface [43,44,45,46]. The two markers were selected based on the 3D structure prediction output. In asexual stages (trophozoites and schizonts), anti-PfSDP antibody signals co-localized with those of anti-MSP-1 antibodies, suggesting association with PfSDP on the merozoite surface (Figure 4A).

With the observation that PfSDP may be associated with the merozoite surface and its likely involvement in cell invasion as determined from its 3D structure prediction, growth inhibition assays were carried out for confirmation purposes using peptide-specific antibodies against *PfSDP*. These antibodies inhibited erythrocyte invasion in a concentration-dependent manner (Figure 4B). Two of the anti-PfSDP peptide-specific antibodies (anti-peptide-2 (PfSDP-2) and anti-peptide-3 (PfSDP-3) inhibited erythrocyte invasion with rates ranging from 6.5% to 50% and 0% to 20% respectively. Anti-peptide-1 (PfSDP-1) did not display any remarkable growth inhibition in 3D7 (Figure 4B). The reference strain 3D7 was more sensitive than the clinical isolate. Pre-immune serum did not inhibit parasite growth. Rather, it tended to potentiate growth.

In sexual stages of the parasite, anti-PfSDP antibodies showed signals beneath Pfs48/45 with partial co-localization at some sections of the parasite (Figure 5). These results suggested that PfSDP localizes partially at the gametocyte body. Moreover, native PfSDP stained as aggregates in gametocytes, implying that this protein may be associated with the membranous organelle.

### 3.5. Plasma Samples from Children and Adults from Different Endemic Areas Reacted to PfSDP Recombinant Proteins

Recombinant PfSDP was expressed in the *E. coli* system and detected on Coomassie blue staining after nickel-nitrilotriacetic acid (Ni-NTA) affinity purification as a dominant species of about 29 kDa (Figure 6A). This was slightly different from the expected size of the native proteins (30.6 kDa) because the sequences coding for signal and transmembrane domains were removed before cloning. The Ni-NTA-purified proteins were concentrated using a 10 kDa cut-off centrifugal filter and subsequently subjected to further purification using size exclusion chromatography using the AKTA pure ^TM^ protein purification system (Figure 6B). The purified recombinant protein was detected by immunoblotting using anti-6xHis monoclonal antibodies (Figure 6C) and pooled plasma samples from malaria-exposed and non-exposed individuals (Figure 6D,E).

To assess the acquired antibody response to PfSDP, a total of 447 plasma samples were used, of which 45 (10%) were from Accra, 155 (34.7%) from Ho, 92 (20.6%) from Kintampo, and 155 (34.7%) from Navrongo. In Navrongo, 49/155 (31.6%) samples were collected from adults. Accra and Ho are known to exhibit low malaria transmission, while Kintampo and Navrongo are known to be high-transmission areas. The description of the study populations is given in Appendix A. Non-immune serum was used as the negative control, and a study participant was considered seropositive when their antibody level was greater than the cut-off value. The cut-off was set to be the mean antibody level of negative controls + 3SD. The PfSDP protein was found to be immunogenic, with seropositivity greater than 80% in all the localities (Figure 6F). Navrongo, which is a high-transmission area, had the highest seroreactivity (95%) in children (0–15 years) with acute malaria. However, this was not the case for Kintampo (the highest transmission site), which registered significantly lower values relative to the other sites. Plasma samples from adults were also found to be highly reactive to PfSDP, with the highest seropositivity (98.5%). Antibody response did not follow the trend of transmission intensity, as Kintampo, with the highest transmission potential, was found with relatively lower antibody levels compared to the other sites (Figure 6G). Children from Navrongo and adults were found to have significantly higher IgG titres compared to children from the other sites (Figure 6G). Adults (>18 years) were enrolled in Navrongo only.

#### 3.5.1. Associations Between Antibody Response Level, Age, Disease Episode, Parasite Density, and Exposure Among the Study Population

To test the effect of age, which is a surrogate of exposure, the study participants were categorized into three groups: <6 years, 7–15 years, and adults (>15 years) (Appendix A). Comparing these groups, there was no significant association of IgG level with age among children (<15 years) irrespective of site (Figure 7A,B). There was a significant increase in IgG response in adults compared to children less than 15 years old (Figure 7A).

To find out whether antibodies raised against the malaria parasite antigen had a protective effect on clinical episodes, parents at enrolment into this study were asked to declare the number of times that their child had suffered from malaria within the year. The information obtained was used to group patients into three groups: a group of children with no clinical episode per year, a group with 1–2 clinical episodes per year, and another group with >3 clinical episodes yearly (Appendix A). Children with less frequent episodes of malaria had significantly higher PfSDP antibody levels compared to those with a high frequency of clinical episodes (*p* < 0.01) when the whole study population was considered in the analysis (Figure 7C). While considering the different study sites, no significant association was found between the antibody response and disease episode (Figure 7D). Samples were also collected from individuals with sub-clinical (asymptomatic) infections in Navrongo. Comparing their IgG levels with those with symptomatic infections revealed a higher IgG level in asymptomatic individuals (*p* < 0.05) in the 7–15 years group but not in children less than 7 years old (Figure 7E,F).

Global analysis revealed a significant negative association between antibody response and parasitemia (*p* < 0.05, Figure 7G). However, this association was only found in Navrongo when the analysis was performed by site (*p* < 0.05, Figure 7H). Globally, there was a weak significant negative correlation between PfSDP antibody titres and parasitemia (r = −0.2, *p* = 0.001). However, this correlation became non-significant within the study site except in Navrongo (r = −0.27, *p* = 0.0044). For the correlation analysis, only seropositive individuals harbouring the parasites were considered in the analysis. A multiple linear regression analysis showed no association between IgG level, age, parasite load, and transmission intensity.

In Navrongo, 6/85 (7.3%) asymptomatic individuals were found to harbour gametocytes in their blood. We sought to check whether there was an association between gametocytemia and IgG titres. Individuals harbouring gametocytes appeared to have slightly lower levels of PfSDP IgG antibodies. Using only gametocyte-positive samples, Spearman rank correlation revealed a strong but non-significant positive relationship between IgG titres and gametocyte PfSDP (r = 0.77, *p* = 0.07).

#### 3.5.2. Time Point-Specific Variations in PfSDP Seropositivity and IgG Level in Malaria-Infected Children Were Observed Across Areas with Different Transmission Potentials

To evaluate the kinetics of PfSDP antigen-specific antibody responses, plasma samples were collected on day 7 and 21 (convalescence) post-treatment in addition to the ones collected on day 0 before treatment (acute infection). IgG level assessment showed time-dependent variations in seroreactivity to PfSDP. The seropositivity increased remarkably from the acute infection (day 0) to day 7 post-treatment, followed by a decrease towards day 21 post-treatment, irrespective of the sample collection site (Figure 8A). The fold increase (seropositivity day 7/seropositivity day 0) in seropositivity of 1.2 was recorded in all three sites (Accra (*p* = 0.004), Ho (*p* < 0.0001), and Kintampo (*p* = 0.001)). Samples were not collected on day 21 in Ho based on the design of that study. The same trend was observed while comparing IgG levels in each child at the different time points (acute infection versus convalescence). Like seropositivity, IgG titres were found to significantly increase from day 0 to day 7, followed by a slight decrease to day 21 for the majority of participants, 94.8% in Accra (Figure 8B), 88.4% in Ho (Figure 8C), and 73.9% in Kintampo (Figure 8D). The remaining participants had their IgG titres either decreasing from day 0 to day 21 or remaining steady throughout the study period. Comparing mean IgG titer per time point showed that IgG titer remained significantly higher at day 21 relative to the baseline day 0 data in Accra but not in Kintampo (Figure 8E,G). The fold increase in antibody titre (mean IgG titre day7/mean IgG titre day0) was 2.5 in Accra (*p* < 0.001), 1.7 in Ho (*p* < 0.001), and 1.5 in Kintampo (*p* < 0.001). Contrasting IgG levels in children from the three sites at different time points showed that children from low-endemic areas (Accra and Ho) reacted more than those from the high-transmission area (Kintampo) at day 7 and day 21 (Figure 8E–G and Appendix A). On day 0, mean antibody levels were similar across the three sites. Follow-up was not performed in Navrongo, hence, the exclusion of this site from the analysis.

## 4. Discussion

This study sought to select and characterize *Plasmodium falciparum* parasite genes to determine their likelihood to be potential vaccine candidates. The selected gene, PF3D7_110580, is highly conserved globally, exhibiting limited and normally distributed genetic diversity across all study countries.

As a preliminary step, the gene was prioritized based on its expression patterns using available *P. falciparum* laboratory strain transcriptome and proteomic data (published and unpublished) and the standard vaccine candidate-associated criteria. Genomic sequence analysis revealed that the gene was largely conserved with the few polymorphisms found under directional selection based on the negative Tajima’s D. Previous studies have found many *P. falciparum* genes to be either under directional or balancing selection [47,48,49,50]. It is worth noting that immune or drug pressure, or both, play an important role in selection [48,50]. This is an indication that the selected gene may be of interest. However, as previously observed with *P. falciparum* parasites, the negative Tajima’s D could also be a result of recent population expansion [51]. Nucleotide and haplotype diversity were found to be higher in Africa than SEA. These findings are in line with a previous study, which revealed higher haplotype diversity of *var2csa* in Africa than SEA [52]. This observation could be attributed to the higher transmission potentials and the older nature of African genomes. An interesting finding in our study is that the selected gene was found to be conserved across continents, and even the limited diversity found was normally distributed in all countries. This was revealed by the heatmap and the PCA scatter plots, showing that the genetic diversity was not linked to any specific geographical regions. Similar results have been obtained with the N-terminal region in circumsporozoite surface protein (CSP), the main antigen in the RTS’S vaccine [53,54]. Moreover, haplotype network analysis revealed the presence of major haplotypes both in Africa and SEA. Taken together, these findings imply that a vaccine containing the relevant variants of these genes could provide strain-transcending immunity.

RT-qPCR analysis showed that PfSDP is expressed across various parasite stages, with its highest expression in early gametocytes. This expression pattern in field isolates matched observations from the 3D7 lab strain, which was used for prioritization in PlasmoDB. The expected expression of the gametocyte-specific Pfs25 positive control indicates that our experimental design was sound and free from contamination by asexual stages. This validates the results obtained from our experiments. This result was confirmed by immunofluorescence and immunoblot experiments that revealed similar expression patterns at the protein level. Based on Western blot data, the PfSDP protein may be polymeric in nature and more likely to undergo stage-specific processing in the parasites. PfSDP exhibited an approximately 37kD band, distinct from its expected monomer and dimer forms, which was predominantly observed in early gametocytes and trophozoites. This suggests that the protein may undergo specific processing for biological functions during these parasite stages. Early gametocyte and trophozoite parasite stages are known to sequester in tissues, but the mechanism involved is not fully elucidated yet, especially in the case of maturing gametocytes [55]. PfSDP structure was modelled from that of a Photosynthetic Reaction Centre Protein whose orthologues have been revealed to be indispensable and likely to be implicated in cell invasion in *Toxoplasma gondi* [39]. Moreover, the structure was predicted to have a subtilisin/kexin isozyme-1 (SKI1) domain, hence, the name *P. falciparum* subtilisin-like domain-containing protein (PfSDP). Subtilisin-like serine proteases in *P. falciparum* are enzymes involved in the pre-processing of proteins, some of which are involved in cell invasion (e.g., MSPs), and, as such, they are considered key regulators of disease pathogenesis [56,57,58]. This is an indication that this protein could have an important role in the parasite during egress and RBC invasion. It is worth mentioning that PfSDP co-localized with MSP-1 at the schizont stage in our study, suggesting that they are both associated with the merozoite surface. MSP-1 is involved in the tight junction formation during the invasion process, supporting the potential involvement of PfSDP in cell invasion.

To complement the structure and function prediction, we also showed that peptide-specific antibodies against PfSDP could mildly inhibit red blood cell invasion in a concentration-dependent manner for both lab and clinical parasite lines. Antibodies to PfSDP peptides had more effects on the lab compared to clinical parasite lines. Parasite lines have previously been shown to respond differently to antibodies against invasion-related genes such as PfEBA181, PfCyRPA, and PfRAMA [59]. The observed low inhibitory activity against erythrocyte invasion may be attributed to the antigen’s potential expression across stages. Some studies involving other invasion-related genes such as RH5 [60,61] have shown that higher antibody concentrations (up to 1 mg/mL or more) than the ones tested in this study (less than 250 ug/mL) exhibit better GIA than observed in the present study. Unfortunately, this could not be achieved here due to limited and less concentrated stock.

PfSDP co-localized partially with a known vaccine candidate, Pfs48/45, in gametocytes. Pfs48/45 plays a role in parasite fertilization in the mosquito midgut [62,63,64,65,66,67]. In the asexual parasite, PfSDP showed membrane localization in late trophozoites/early schizonts but showed a peri-nuclear staining pattern like PfMSP-1 in mature schizonts [68,69,70]. The proximity of PfSDP to known vaccine candidates suggested that it is likely associated with the membrane and exposed to the human host immune system. In that context, we went further to evaluate the naturally acquired antibody responses to this antigen in children and adults living in different endemic areas.

The antigen was found to be immunogenic, with antibody levels for most of the children and adults above the cut-off. The samples were collected from malaria-endemic areas with different transmission intensities, with Kintampo having the highest entomological inoculation rate >250 infective bites/person per year (ib/p/yr), followed by Navrongo with <250 ib/p/yr, Ho with 69 ib/p/yr, and Accra with <50 ib/p/yr as previously reported [71,72,73,74]. Analysis based on the site failed to show a relationship between transmission intensity and neither antibody level nor seropositivity. This observation was previously made with some *P. falciparum* invasion-related merozoite proteins, such as MSP-7, PF3D7_1404900, PF3D7_038300, Armadillo-Type Repeat Protein (PfATRP) [75], reticulocyte-binding homologue antigen RH4.2, and erythrocyte-binding antigen 140 (EBA140) [76]. This suggests that this antigen cannot be used as a biomarker for disease transmission intensity. Kintampo, a high-transmission site, registered lower IgG titres [77].

Despite the significantly higher increase in IgG level in adults, age was not found to significantly influence antibody response in children less than 15 years old [75]. This suggests that these antigens may need more exposure time to boost the immune system. Globally, there was a significant negative but weak association between IgG titre and parasite load, which was reproducible only in Navrongo. Although no significant association was found between gametocytemia and IgG levels in the human host, it would be interesting to evaluate the transmission-blocking activity of these antibodies in the vector using a standard membrane feeding approach [63,66,67,78].

A major issue with common malaria vaccine candidates is that they elicit the production of short-lived antibodies or memory B cells. To address that issue in this study, samples were collected during acute infection (day 0) and convalescence (day 7 and day 21). Both the seroreactivity of children to the antigen and the level of specific antibody response were found to markedly increase from day 0 to day 7, followed by a drop on day 21. B cell populations are known to decline in the absence of the stimulating antigen via cellular processes such as programmed cell death, with the subsequent production of memory cells [79]. This might explain the drop in the IgG levels observed in this study on Day 21. Though B cell populations were not evaluated in this study to ascertain this claim, the findings are consistent with previous data reported on *P. falciparum* invasion-related antigens (AMA-1, MSP-119, MSP-2, MSP-3, and RH5), which have been shown to stimulate the production of short-lived antibodies [79,80,81,82]. A similar observation was made for antibody response to PfRh5, which decayed rapidly within 42 days post-treatment after a sharp increase from disease onset in Ghanaian children. Also, AMA-1 antibody levels and specific B cell counts were shown to decrease from convalescence to the 12th month in Malian children [79]. This is an indication that the levels of total IgG against our selected antigen, PfSDP, could constitute markers of recent *P. falciparum* exposure. However, it is worth mentioning that IgG levels in Ho were still significantly higher on day 21 compared to the day 0 baseline level despite the drop. Memory B cells in concert with T cells are important in the presence of the antigens. A good vaccine does not only rely on how protective antibodies are but also on the persistence of its specific memory B cells [81]. Studies geared towards evaluating the breadth and kinetics of specific memory B cells and T cells are highly warranted to complement the findings of the present study.

## 5. Conclusions

In this study, *PfSDP* was found to be relatively conserved across continents and likely under directional selection. It displayed cross-stage expressions both at the transcript and protein level in clinical isolates. The protein is more likely to be associated with the gametocyte body and merozoite surface in asexual stages. Structure and motif predictions suggested that they might be involved in cell invasion-related processes. Peptides specific to PfSDP inhibited erythrocyte invasion to some extent. The protein is immunogenic in children from different endemic areas with both seroreactivity and antibody levels increasing from acute infection to day 7 post-treatment, followed by a gradual decay.

## Figures and Tables

**Figure 1 cells-14-01184-f001:**
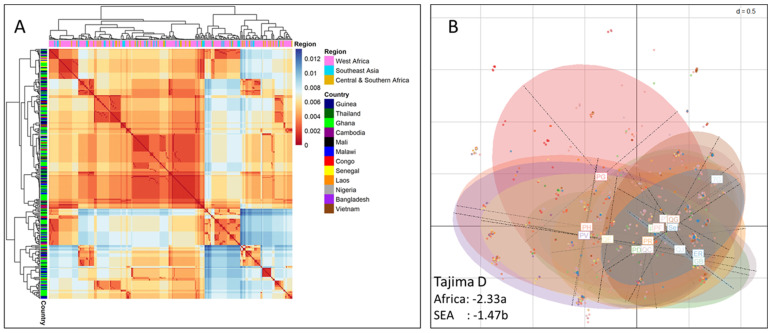
Heatmap and principal component analysis (PCA) scatter plot summarizing the genetic diversity per geographical location of *P. falciparum* subtilisin-like domain containing protein (*pfsdp*, Pf3D7_1105800) gene: (**A**) Gene sequences are aligned using Mafft_auto and used for the sequential and parallel maximum likelihood-based inference of phylogenetic trees, which is finally used for the generation of phylogenetic trees and heatmaps in the R-software (v3.4). (**B**) Allele counts (frequencies) are extracted from the VCF, cleaned, and then used to perform the PCA. Estimates of selection indices are calculated. The departure from neutrality is more pronounced in Africa than in SEA. Squares at the centre of ellipses represent the different countries. The data used is from 12 countries: 7 in Africa (Congo, Nigeria, Malawi, Mali, Ghana, Senegal, and Guinea) and 5 in South-East Asia (Laos, Vietnam, Bangladesh, Cambodia, and Thailand) [PFK3 release 1]. a = significant, b = non-significant.

**Figure 2 cells-14-01184-f002:**
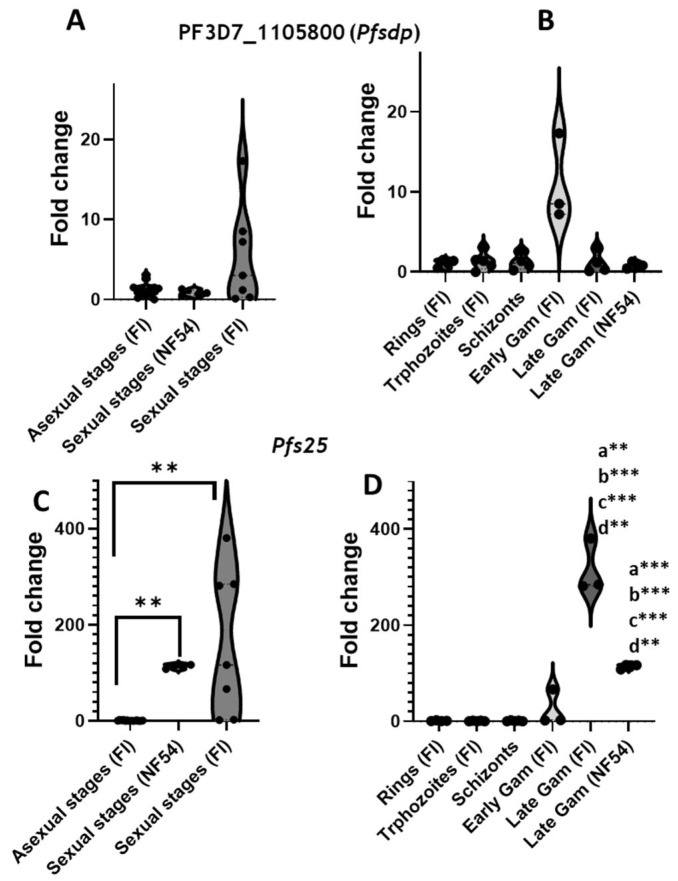
Assessment of *pfsdp* (Pf3D7_1105800) expression in clinical isolates (FI). Differential expression of the *pfsdp* gene is consistent with observations in laboratory strains (PlasmoDB). The gene has cross-stage expression as shown by RT-qPCR using cultivated field isolated parasites in comparison with the NF54 lab strain as a control (**A**,**B**). Pfs25, a gametocyte-specific gene, is used as a positive control to ascertain the purity of the samples and is upregulated in late gametocytes with very low expression in asexual stages (**C**,**D**). *p* value code: **: *p* ≤ 0.01, ***: *p* ≤ 0.001. a, b, c, and d indicate significance upon comparing late gametocyte gene expression level to that of rings, trophozoites, schizonts, and early gametocytes, respectively.

**Figure 3 cells-14-01184-f003:**
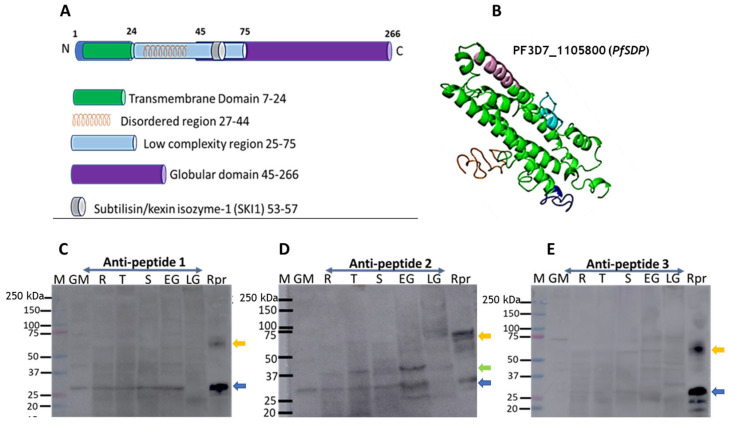
Assessment of PfSDP (Pf3D7_1105800) protein 3D structure prediction and expression: (**A**) Topographical representation of the different domains of the PfSDP protein. (**B**) Predicted 3-dimensional structure. Structure prediction is performed using I-TASSER and visualized in PyMol. Topographical representation is generated using I-TASSER and Eukaryotic linear domain (ELM) resources. (**C**–**E**) Immunoblots with lysates derived from different developmental stages (rings (R), trophozoites (T), schizonts (S), and early (EG) and late (LG) gametocytes) of *P. falciparum* while using recombinant protein as a control (Rpr). M: Molecular weight markers; GM: Ghost membrane; Rpr: recombinant protein; blue arrow: monomer; green arrow: potential processed or interacting proteins; and orange arrow: dimer.

**Figure 4 cells-14-01184-f004:**
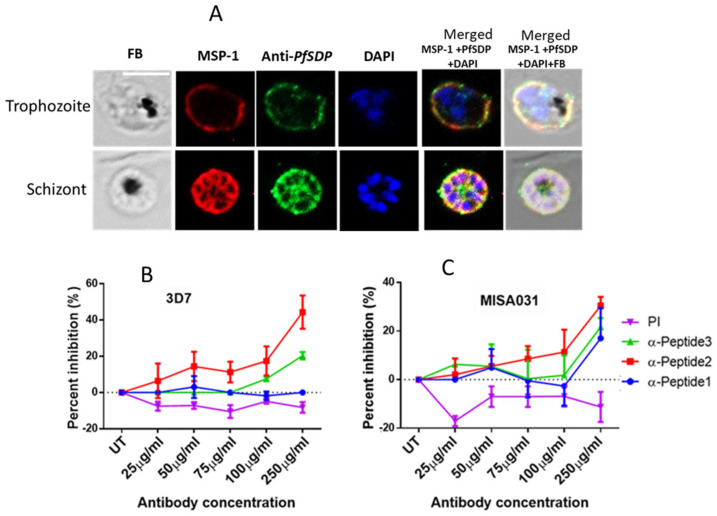
Asexual stage parasite localization of the native PfSDP (Pf3D7_1105800) protein and growth inhibition assays: (**A**) Immunofluorescence assays (IFAs). Parasites are co-stained with anti-PF3D7_1105800 (anti-PfSDP) anti-Peptide-1, and a merozoite surface (MSP-1) marker antibody. Secondary antibodies used are Alexa Fluor 488-conjugated goat α-mouse immunoglobulin G (IgG) for PfSDP (green) and Alexa Fluor 568-conjugated goat α-rabbit IgG for the marker MSP-1. Parasite nuclei are stained with DAPI (blue). The differential interference contrast (DIC) and merged images are shown. (**B**,**C**) Growth inhibition assays (GIAs) with antibodies raised in rabbit against PfSDP-specific B cell epitopes (short peptides: PfSDP-1, PfSDP-2, and PfSDP-3) using a lab strain 3D7 and a clinical isolate MISA031 respectively. All three peptide-specific antibodies inhibited red blood cell invasion in a concentration-dependent manner, with α-PfSDP-1 antibodies showing a limited effect solely against the clinical isolate. The clinical isolate parasite line, MISA031 (**C**), and lab strain 3D7 (**B**) were cultured with different concentrations of antibodies. Rabbit pre-immune (PI) serum is used as a negative control. Parasitemia is determined by flow cytometry. Two independent assays are carried out in duplicates. The MISA031 parasite line is also used for IFAs. Scale bar on IFA pictures represents 5 μm, while the error bars on GIA pictures represent the standard errors of the means.

**Figure 5 cells-14-01184-f005:**
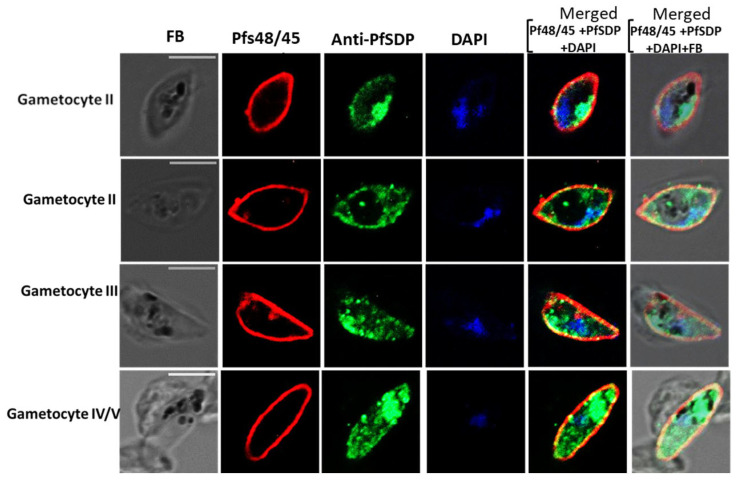
Sexual stage parasite localization of the native PfSDP (Pf3D7_1105800) protein using IFA. Gametocytes are co-stained with anti-PF3D7_1105800 (anti-PFSDP) and gametocyte surface (Pfs48/45) marker antibodies. Secondary antibodies used ae Alexa Fluor 488-conjugated goat α-mouse immunoglobulin G (IgG) for PfSDP (green) and Alexa Fluor 568-conjugated goat α-rabbit IgG for the markers MSP-1 and Pfs48/45 (red). Parasite nuclei are stained with DAPI (blue). The differential interference contrast (DIC) and merged images are shown. The MISA031 parasite line is also used for IFAs. The scale bar represents 5 μm.

**Figure 6 cells-14-01184-f006:**
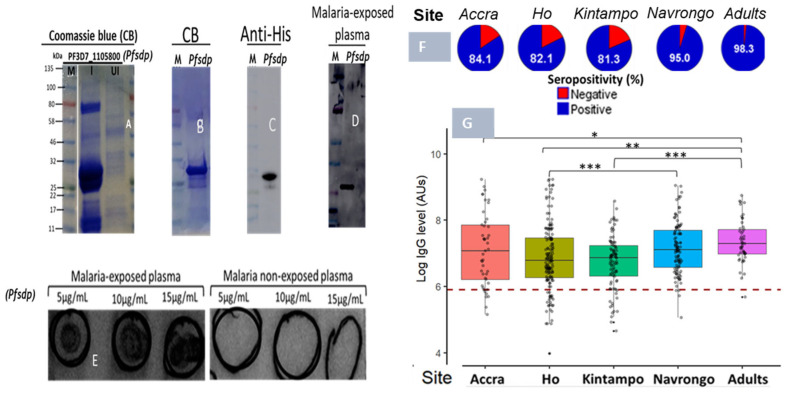
Production of recombinant PfSDP (Pf3D7_1105800) and reactivity to naturally acquired antibodies: (**A**) The crude extracts of IPTG-induced (I) and un-induced (UI) *E. Coli* cultures are subjected to SDS-PAGE and stained with Coomassie Brilliant Blue (CBB). The target band of about 29 kDa is detected as the dominant species in the induced sample. (**B**) CBB stain after size exclusion chromatography. (**C**) Western blot with anti-histidine HRP-conjugated antibodies (1:10,000). (**D**,**E**) Naturally acquired antibodies recognized PfSDP recombinant proteins; Western blot using plasma from malaria-infected patients. (**D**) Dot blots with plasma from malaria-infected and non-infected patients. (**E**) For dot blot, 10 µL of the recombinant protein at the indicated concentrations (5, 10, and 15 µg/mL) is applied onto a nitrocellulose membrane. (**F**) Seropositivity of malaria-exposed children and adults to the PfSDP recombinant protein in different endemic areas. A study participant is considered seropositive when their Day 0 antibody level is greater than the cut-off value ----. Cut-off = mean antibody level of the naive European plasma + 3SD. (**G**) Immunogenicity levels across sites of contrasting endemicities: Accra (*n* = 45), Ho (*n* = 155), Kintampo (*n* = 92), and Navrongo (n = 155 (106 children and 49 adults)). Accra and Ho are low-endemic areas, while Kintampo and Navrongo are high-endemic areas. *p* value code: ns: *p* > 0.05. *: *p* ≤ 0.05, **: *p* ≤ 0.01, ***: *p* ≤ 0.001. IPTG: Isopropyl β-D-1-thiogalactopyranoside; M: molecular marker.

**Figure 7 cells-14-01184-f007:**
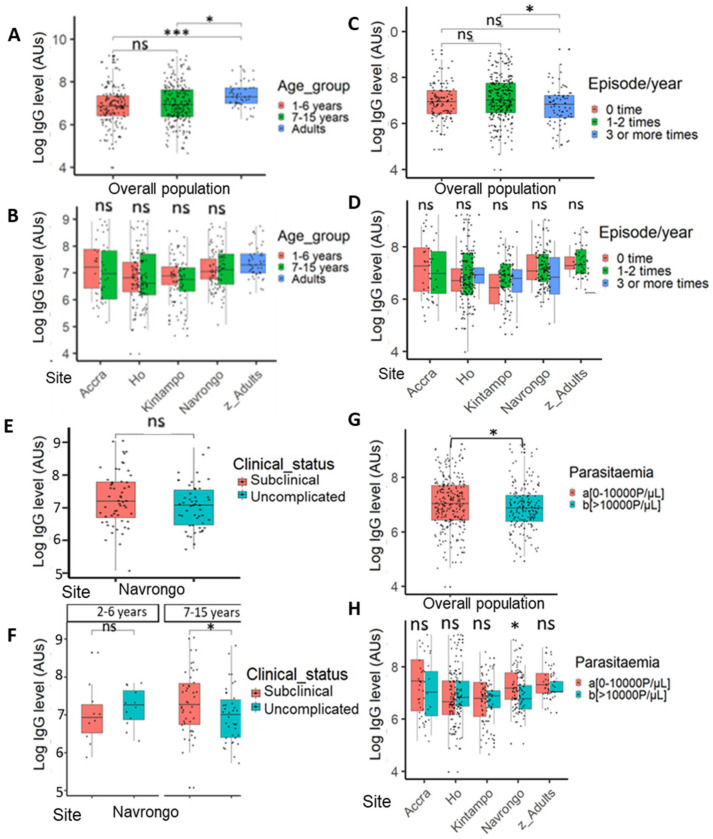
Associations between PfSDP antibody response level, age, disease episode, and parasite density among the study population: (**A**) IgG level by age with the overall population; (**B**) IgG level by age and site; significant increase in IgG response in adults compared to children less than 15 years old. Slight decrease in IgG level with age in children less than 15 years (**A**). (**C**) Association between IgG level and disease episode in the overall population. (**D**) Association between IgG level and disease episodes per site. A globally significant negative association between IgG levels and disease episodes per year, which disappears when sites are considered. (**E**) Association between IgG level and clinical status of malaria in Navrongo children. (**F**) Association between IgG titre and clinical status of malaria in Navrongo by age. (**G**) Global association between IgG levels and parasite load. (**H**) Association between IgG level and parasite load per site. Globally significant negative association between IgG levels and parasite density, which disappears when sites are considered, except in Navrongo (high-endemic area). Mean IgG levels are compared using Kruskal–Wallis (KW) and Mann–Whitney tests. When a KW test is significant (*p*-value < 0.05), a Dunnett’s test is used as post hoc for multiple comparisons. *p*-value code: ns: *p* > 0.05. *: *p* ≤ 0.05, ***: *p* ≤ 0.001.

**Figure 8 cells-14-01184-f008:**
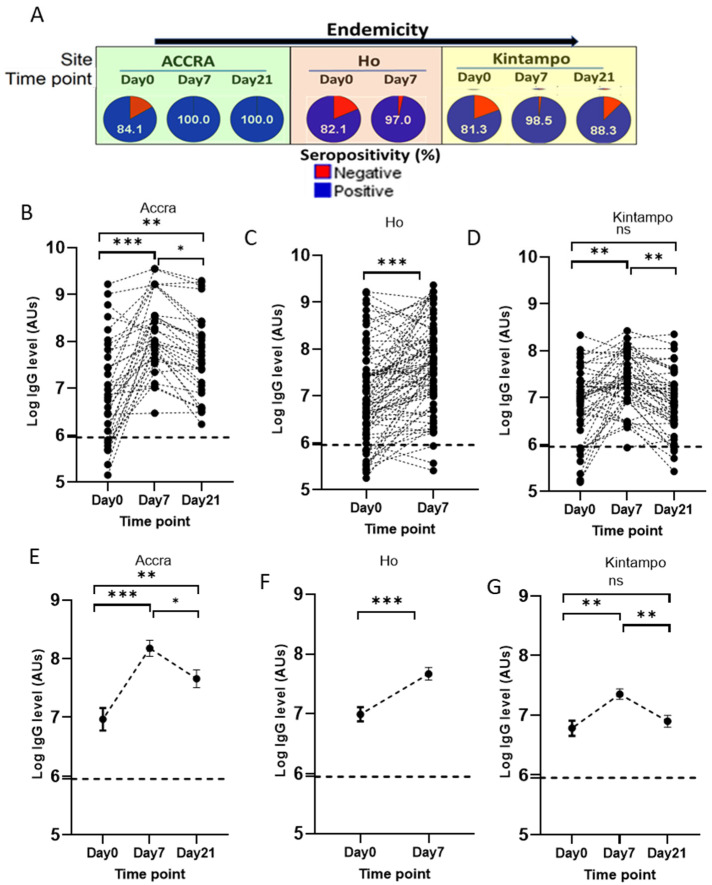
Kinetics of anti-PfSDP specific antibody response in malaria-infected children from acute infection (day 0) to convalescence (Day 7 through day 21) in areas with different transmission potentials: (**A**) Variation in seropositivity in children at different time points. Kinetics of anti-PfSDP specific antibody response per individual in (**B**) Accra, (**C**) Ho, and (**D**) Kintampo. Mean comparison between time points in (**E**) Accra, (**F**) Ho, and (**G**) Kintampo. An increasing trend of antibody levels is observed from day 0 to day 7, followed by a gradual decrease till day 21. Follow-up is not performed in Ho on Day 21. Cut-off ------- = the mean ab level of naïve European plasma + 3SD. Mean IgG levels were compared using Kruskal–Wallis (KW) and Mann–Whitney tests. When a KW test was significant (*p*-value < 0.05), a Dunnett’s test was used as a post hoc test for multiple comparisons. *p*-value code: ns: *p* > 0.05. *: *p* ≤ 0.05, **: *p* ≤ 0.01, ***: *p* ≤ 0.001.

## Data Availability

The original contributions presented in this study are included in this article/Appendix A. Further inquiries can be directed to the corresponding authors.

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
