# Peer review of "Plasmodium falciparum Subtilisin-like Domain-Containing Protein (PfSDP), a Cross-Stage Antigen, Elicits Short-Lived Antibody Response Following Natural Infection with Plasmodium falciparum"

_cells, 2025, doi:10.3390/cells14151184_

Round 1
Reviewer 1 Report
Comments and Suggestions for Authors
This manuscript by Kengne-Ouafo et al. shows an important aspect of malaria vaccine development by characterizing a novel P. falciparum Subtilisin-like Domain containing Protein (PfSDP) expressed across parasite stages. Specifically, this study demonstrates the potential of PfSDP as a vaccine candidate by elucidating the gene’s expression throughout the parasite’s sexual and asexual stages, its co-localization with other key proteins, and the inhibitory effects of anti-PfSDP antibodies on erythrocyte invasion. However, I recommend that the manuscript be strengthened and revised with consideration of the following aspects.
1. The reported erythrocyte invasion inhibition by PfSDP-specific antibodies is modest (ranging from 6.5–50% for one peptide and 0–20% for another) with a broad variability. In addition, one of the peptides (peptide 1) shows negligible inhibition. It is advisable to repeat these assays under more controlled and varied conditions (e.g., a wider range of antibody concentrations, different incubation times) and consider comparing the results with antibodies raised against the full-length protein to evaluate reproducibility and functional significance.
2. The antibody response shows a significant increase at day 7 post-infection followed by a rapid decline by day 21, yet the underlying mechanism is not discussed in depth. Memory B-cell responses were not assessed, which raises concerns about the durability of the immune response. The authors should consider repeating experiments to include a memory B-cell assay or provide additional supporting data from the literature to explain the observed transient antibody response.
3. While a global analysis showed weak or inconsistent correlations between antibody levels, parasite density, and clinical episodes, the data are not robust across all study regions. The authors might need to perform multivariate regression analyses such as age, geographic location, and baseline immunity to strengthen or clarify the observed associations.
4. The localization studies show partial co-localization of PfSDP with MSP-1 and Pfs48/45, but the data are presented qualitatively without quantitative co-localization metrics. It would be beneficial to repeat the IFA experiments and include a quantitative analysis of co-localization to substantiate the claims regarding the spatial distribution of PfSDP in relation to established markers.
5. The manuscript does not provide sufficient details on whether the selected three peptides adequately represent the immunologically relevant regions of the full-length PfSDP protein. Additional structural data or sequence alignments should be provided (or experiments repeated) to confirm that these peptides are indeed representative of key domains that drive the immunogenicity and biological function of the antigen.
Author Response
This manuscript by Kengne-Ouafo et al. shows an important aspect of malaria vaccine development by characterizing a novel P. falciparum Subtilisin-like Domain containing Protein (PfSDP) expressed across parasite stages. Specifically, this study demonstrates the potential of PfSDP as a vaccine candidate by elucidating the gene’s expression throughout the parasite’s sexual and asexual stages, its co-localization with other key proteins, and the inhibitory effects of anti-PfSDP antibodies on erythrocyte invasion. However, I recommend that the manuscript be strengthened and revised with consideration of the following aspects.
1. The reported erythrocyte invasion inhibition by PfSDP-specific antibodies is modest (ranging from 6.5–50% for one peptide and 0–20% for another) with a broad variability. In addition, one of the peptides (peptide 1) shows negligible inhibition. It is advisable to repeat these assays under more controlled and varied conditions (e.g., a wider range of antibody concentrations, different incubation times) and consider comparing the results with antibodies raised against the full-length protein to evaluate reproducibility and functional significance.
Response: The antibodies used in the experiment were raised against synthetic peptides of the antigen. The purified antibody levels could only allow this range of dilutions to be evaluated. The different invasion inhibitory potential varies as the peptides were from different parts of the protein. Over the years of conducting invasion inhibitory assays, over-saturating the system with more antibodies tends to cause some irregularities. Once we are within the linear performance of the antibodies, we think it is okay. The assay is very reproducible, as this was repeated independently three times, and in each case, the assay was conducted in triplicate. Initially, expressing the full-length protein was challenging, hence the need to resort to peptides. We could later express the protein but at very low amounts.
2. The antibody response shows a significant increase at day 7 post-infection followed by a rapid decline by day 21, yet the underlying mechanism is not discussed in depth. Memory B-cell responses were not assessed, which raises concerns about the durability of the immune response. The authors should consider repeating experiments to include a memory B-cell assay or provide additional supporting data from the literature to explain the observed transient antibody response.
Response: We are grateful to the reviewer for this comment. A substantial paragraph has been added to the discussion on the underlying mechanism of the IgG decline on Day 21. We have also mentioned the lack of b cell assays as a limitation and recommended that for further studies.
3. While a global analysis showed weak or inconsistent correlations between antibody levels, parasite density, and clinical episodes, the data are not robust across all study regions. The authors might need to perform multivariate regression analyses such as age, geographic location, and baseline immunity to strengthen or clarify the observed associations.
Response: This analysis was done but revealed no significant impact of age, parasite load, and transmission intensity on IgG response. This has been mentioned in the revised manuscript.
The localization studies show partial co-localization of PfSDP with MSP-1 and Pfs48/45, but the data are presented qualitatively without quantitative co-localization metrics. It would be beneficial to repeat the IFA experiments and include a quantitative analysis of co-localization to substantiate the claims regarding the spatial distribution of PfSDP in relation to established markers.
Response: The co-localization is partial. At the resolution we are working with, performing co-localization analysis will be greatly misleading and over-interpretation of the data. Generally, the observed localization and staining pattern is necessary in helping ascribe a possible function to the protein. The experiments have been repeated at least 6 times.
5. The manuscript does not provide sufficient details on whether the selected three peptides adequately represent the immunologically relevant regions of the full-length PfSDP protein. Additional structural data or sequence alignments should be provided (or experiments repeated) to confirm that these peptides are indeed representative of key domains that drive the immunogenicity and biological function of the antigen.
Responses: the selected peptides were scored for their Antigenicity/Surface/Hydrophilicity, disorder scoring. Also factored into the selection is consideration of peptides towards the N-, Middle, and C-termini of the protein. Since the protein lacked classical domains, this criterion will allow us to know the different functional relevance of the different regions of the protein.
Reviewer 2 Report
Comments and Suggestions for Authors
With the increasing emergence of artemisinin resistance to front-line antimalarials in Africa the search for novel, highly efficacious malaria vaccines remains a critical priority. In the manuscript, the authors performed an in silico screening to identify Plasmodium falciparum gametocyte-stage genes with potential as targets for protective immunity or diagnostic markers. Through this analysis, they identified the gene Pf3D7_1105800, which encodes a subtilisin-like domain-containing protein (PfSDP). The authors not only demonstrated both transcript and protein-level expression of the protein across all developmental stages of the parasite, but also its co-localization with merozoite surface protein 1 (MSP-1) and the gametocyte surface marker Pfs48/45. Importantly, antibodies raised against specific PfSDP-derived peptides significantly inhibited erythrocyte invasion in a dose-dependent manner, suggesting a possible role for PfSDP in merozoite invasion. Additionally, a population-based study demonstrated that PfSDP is immunogenic in children from different malaria-endemic regions. Together, these findings support the selection and characterization of PfSDP as a promising candidate for further evaluation in the development of a new P. falciparum vaccine.
However, the manuscript would benefit from revisions to enhance clarity, grammar, and consistency.
Line 2: It is recommended to remove the gene identification code from the title of the manuscript (Pf3D7_1105800).
Line 42: Check that the page (https://www.malariavaccine.org/malaria-and-vaccines/rtss) can be accessed. It seems to be down. If it doesn't work, replace the reference.
In the Introduction section, providing more detailed information on the current efficacy rates of available vaccines and existing treatment options would enhance the reader’s understanding of the relevance of the research presented in this manuscript.
Several of the references cited should be updated with more recent sources to ensure the research remains relevant and current.
Throughout the manuscript, some references are incorrectly embedded in the text. It is recommended to always use reference management tools and verify their proper functionality.
Line 155: It is recommended to add a new subsection before detailing the immunoblotting methodology. Additionally, the section describing immunoblotting should be placed after the generation of specific antibodies (2.5), as these antibodies were used for immunoblotting.
It is recommended to carefully review the manuscript. Some sentences contain closing parentheses without corresponding opening ones, missing periods, and inconsistent capitalization.
Line 282: Provide appropriate references for the Western Blot and Dot Blot techniques.
Line 306: Section 3.1: Gene Selection Criteria, does not describe a result but rather the criteria used. It is recommended that this information be included only in the Materials and Methods section or that the results obtained from this analysis be described in more detail. Mentioning this information at the beginning of the following section could help improve the reader's understanding.
Improve the image quality of all tables and figures.
Figure 2. It is recommended to describe the left and right panels more effectively. Perhaps adding the labels (A), (B)… to each section of the figure could facilitate its understanding. Describing the meaning of (a)**, (b)***… could enhance the comprehension of the information presented in the graph.
Line 105: report the total number of patient samples used.
Line 180-181: “The three peptides were selected based on predicted antigenicity, surface exposure, and hydrophobicity scores…” It is recommended to mention at least which region of the protein these peptides correspond to.
Please ensure that the supplementary information is uploaded to the platform, as it could not be viewed.
Figure 3. It is suggested that the authors include a loading control for the Immunoblots shown in C, D & E, or at least an image of the stained membranes using Ponceau red.
Figure 4-A: It is recommended that the authors present only one representative image of the schizont that reflects the reported results, as there are no significant differences between lines 2 and 3 in panel A.
Additionally, why is only the immunofluorescence with the antibody against peptide 1 shown, when line 427 mentions the use of antibodies targeting different regions of the protein?
Figure 4 and 5 description: Clarify which samples or strains were used for the immunofluorescence assays.
Line 512: Clarify the cut-off value or remove the parentheses.
Figure 7: Correct the Y-axis labels, as some numbers or descriptions were lost during figure preparation.
Line 592: Clarify "Cut-off (-------)" or remove the parentheses.
In the Discussion section, references should be cited using the appropriate numbering within the text.
Add a list of abbreviations at the end of the manuscript.
Some sentences in the Discussion could be rephrased to avoid redundant terms or explanations. It is recommended that the authors revise the manuscript for clarity and conciseness.
Author Response
With the increasing emergence of artemisinin resistance to front-line antimalarials in Africa the search for novel, highly efficacious malaria vaccines remains a critical priority. In the manuscript, the authors performed an in silico screening to identify Plasmodium falciparum gametocyte-stage genes with potential as targets for protective immunity or diagnostic markers. Through this analysis, they identified the gene Pf3D7_1105800, which encodes a subtilisin-like domain-containing protein (PfSDP). The authors not only demonstrated both transcript and protein-level expression of the protein across all developmental stages of the parasite, but also its co-localization with merozoite surface protein 1 (MSP-1) and the gametocyte surface marker Pfs48/45. Importantly, antibodies raised against specific PfSDP-derived peptides significantly inhibited erythrocyte invasion in a dose-dependent manner, suggesting a possible role for PfSDP in merozoite invasion. Additionally, a population-based study demonstrated that PfSDP is immunogenic in children from different malaria-endemic regions. Together, these findings support the selection and characterization of PfSDP as a promising candidate for further evaluation in the development of a new P. falciparum vaccine.
However, the manuscript would benefit from revisions to enhance clarity, grammar, and consistency.
Response: Thank you for your comments. We have revised the manuscript extensively and believe your concerns have been duly addressed.
Line 2: It is recommended to remove the gene identification code from the title of the manuscript (Pf3D7_1105800).
Response: The gene ID Pf3D7_1105800 has been removed from the title as suggested by the reviewer.
Line 42: Check that the page (https://www.malariavaccine.org/malaria-and-vaccines/rtss) can be accessed. It seems to be down. If it doesn't work, replace the reference.
Response: The link has been replaced by the following : (https://www.malariavaccine.org/existing-vaccines/rts)
In the Introduction section, providing more detailed information on the current efficacy rates of available vaccines and existing treatment options would enhance the reader’s understanding of the relevance of the research presented in this manuscript.
Response: More detailed information on the current efficacy rates of available vaccines and existing treatment options has been provided as suggested by the reviewer, see lines 42-52.
Several of the references cited should be updated with more recent sources to ensure the research remains relevant and current.
Response: Recent references have been added to the introduction and discussion sections
Throughout the manuscript, some references are incorrectly embedded in the text. It is recommended to always use reference management tools and verify their proper functionality.
Response; A reference management tool has been used for more adequate referencing
Line 155: It is recommended to add a new subsection before detailing the immunoblotting methodology. Additionally, the section describing immunoblotting should be placed after the generation of specific antibodies (2.5), as these antibodies were used for immunoblotting.
Response: A subsection has been created for the immunoblotting methodology after the generation of specific antibodies (2.5) as suggested by the reviewer.
It is recommended to carefully review the manuscript. Some sentences contain closing parentheses without corresponding opening ones, missing periods, and inconsistent capitalization.
Response: The manuscript has been proofread for all authors and the necessary changes were made.
Line 282: Provide appropriate references for the Western Blot and Dot Blot techniques.
Response: A reference has been provided for the Western Blot and Dot Blot techniques as suggested
Line 306: Section 3.1: Gene Selection Criteria, does not describe a result but rather the criteria used. It is recommended that this information be included only in the Materials and Methods section or that the results obtained from this analysis be described in more detail. Mentioning this information at the beginning of the following section could help improve the reader's understanding.
Response: Section 3.1 has been removed as the Materials and Methods section already have a similar section.
Improve the image quality of all tables and figures.
Response: The quality of figures has been improved. Some figures have been structured or re-plotted
Figure 2. It is recommended to describe the left and right panels more effectively. Perhaps adding the labels (A), (B)… to each section of the figure could facilitate its understanding. Describing the meaning of (a)**, (b)***… could enhance the comprehension of the information presented in the graph.
Response: Figure 2 has been adjusted following the reviewer’s recommendations.
Line 105: report the total number of patient samples used.
Response: We thank the reviewer for this suggestion. We opted not to present results in the methods section. The total number of patient samples used is given in the results under “PfSDP has Cross-Stage Expression at the Transcript and Protein Levels in Clinical Isolates”
Line 180-181: “The three peptides were selected based on predicted antigenicity, surface exposure, and hydrophobicity scores…” It is recommended to mention at least which region of the protein these peptides correspond to.
Response: The regions of the protein the peptides correspond to have been provided, please, see lines 192-193 and suppl table 1.
Please ensure that the supplementary information is uploaded to the platform, as it could not be viewed.
Response: The supplementary information has been uploaded to the platform. Our sincere apology for this omission in the initial submission.
Figure 3. It is suggested that the authors include a loading control for the Immunoblots shown in C, D & E, or at least an image of the stained membranes using Ponceau red.
Response: Ponceau Red staining was not performed but we provided internal experimental controls such as the use of ghost membrane proteins to confirm the lack of cross-reactivity with other host proteins.
Figure 4-A: It is recommended that the authors present only one representative image of the schizont that reflects the reported results, as there are no significant differences between lines 2 and 3 in panel A.
Response: Figure 4 has been adjusted following the reviewer’s suggestion
Additionally, why is only the immunofluorescence with the antibody against peptide 1 shown, when line 427 mentions the use of antibodies targeting different regions of the protein?
Response: Antibody against peptide 1 gave better staining than 2 and 3. We decided to present the results with the anti-peptide antibodies only. Moreover, we ran short of anti-peptide 2 and 3 as they were all used for growth inhibition assays
Figure 4 and 5 description: Clarify which samples or strains were used for the immunofluorescence assays.
Response: We have updated the legends for Figures 4 and 5 (lines 465 and 578) to specify that the immunofluorescence assays were conducted using the MISA031 parasite line.
Line 512: Clarify the cut-off value or remove the parentheses.
Response: The parentheses have been removed
Figure 7: Correct the Y-axis labels, as some numbers or descriptions were lost during figure preparation.
Response: The quality of Figure 7 has been improved, and the y-axis labels have been corrected.
Line 592: Clarify "Cut-off (-------)" or remove the parentheses.
Response: The parentheses have been removed
In the Discussion section, references should be cited using the appropriate numbering within the text.
Response: The references have adjusted accordingly
Add a list of abbreviations at the end of the manuscript.
Response: A list of abbreviations has been added at the end of the revised version of the manuscript.
Some sentences in the Discussion could be rephrased to avoid redundant terms or explanations. It is recommended that the authors revise the manuscript for clarity and conciseness.
Response: The discussion has been revised according to the reviewer’s suggestion
Reviewer 3 Report
Comments and Suggestions for Authors
This manuscript presents a combination of basic and clinical research and is of high impact to scientists working in both fields. The manuscript is excellently written and organized. It characterizes a P. falciparum protein with a subtilisin -like domain (PfSDP) and investigates its antigenic function and protection in P.falciparum infection. The protein seems be involved in erythrocyte invasion since it colocalizes with the MSP protein. The protein is immunogenic in children from different endemic regions. However, antibody levels decrease after 7 days post treatment.
I have some questions back to the authors.
- In Fig.4 growth inhibition is shown with different peptide antibodies directed against epitopes of the protein. From the graph in Fig.4, I conclude that peptide 1 does not show growth inhibition. However, in the legend it is mentioned that all peptides show growth inhibition in a concentration dependent manner. How do the authors explain this comment?
- Fig.7 is hard to read in particular part B.
- Fig.8 is not clear in presentation. It is confusing to follow the decay between day7 and day21. This has to be clarified.
There are some typos in the manuscript.
- One bracket too much in the heading.
- line 42 "efficacy remains". I assume that there is a word missing.
- line 48 ..."but continue their development in the mosquito after taking up an infected blood" what Nisan infected blood??
- line 106 Clinical P. falciparum isolates and NF54 reference strain used in this study. Please leave out were...
Author Response
This manuscript presents a combination of basic and clinical research and is of high impact to scientists working in both fields. The manuscript is excellently written and organized. It characterizes a P. falciparum protein with a subtilisin -like domain (PfSDP) and investigates its antigenic function and protection in P.falciparum infection. The protein seems be involved in erythrocyte invasion since it colocalizes with the MSP protein. The protein is immunogenic in children from different endemic regions. However, antibody levels decrease after 7 days post treatment.
I have some questions back to the authors.
- In Fig.4 growth inhibition is shown with different peptide antibodies directed against epitopes of the protein. From the graph in Fig.4, I conclude that peptide 1 does not show growth inhibition. However, in the legend it is mentioned that all peptides show growth inhibition in a concentration dependent manner. How do the authors explain this comment?
Response: We appreciate the reviewer's comment. While anti-peptide 1 exhibited no significant inhibitory effect on the laboratory strain 3D7, it demonstrated minor growth inhibition of the clinical isolate MISA031 at the highest tested concentration (250 µg/mL), as depicted in Figure 4. To reflect this nuance, the legend has been revised (lines 452-454) to state: "All three peptide-specific antibodies inhibited red blood cell invasion in a concentration-dependent manner, with α-PfSDP-1 showing a limited effect solely against the clinical isolate."
- Fig.7 is hard to read in particular part B.
Response: Figure 7 has been structured to make it more comprehensive
- Fig.8 is not clear in presentation. It is confusing to follow the decay between day7 and day21. This has to be clarified.
Response: Figure 7 has been updated with additional descriptive text (lines 594-600). Furthermore, Figure 8 now includes new graphs illustrating the variation in mean IgG response across different time points, clearly showing an increase on day 7 followed by subsequent decay.
There are some typos in the manuscript.
- One bracket too much in the heading.
Response: The extra bracket has been removed.
- line 42 "efficacy remains". I assume that there is a word missing.
Response: The sentence has been adjusted.
- line 48 ..."but continue their development in the mosquito after taking up an infected blood" what Nisan infected blood??
Response: The sentence has been rephrased and read thus. "but continue their development in the mosquito following a gametocyte-infected blood meal”
- line 106 Clinical P. falciparum isolates and NF54 reference strain used in this study. Please leave out were...
Response: The sentence has been revised.
Round 2
Reviewer 1 Report
Comments and Suggestions for Authors
The revised manuscript sufficiently addresses all the comments raised during the first round of peer review. Compared to the original version, the current manuscript has been significantly strengthened both scientifically and technically. However, some minor but important corrections are still required. Specifically, in Figure 3, "kD" should be corrected to "kDa" to accurately reflect the standard unit of molecular weight. Additionally, Figure 4 and Figure 5 lack scale bar measurements, which are essential for proper interpretation of immunofluorescence data. These issues must be corrected.
Author Response
The revised manuscript sufficiently addresses all the comments raised during the first round of peer review. Compared to the original version, the current manuscript has been significantly strengthened both scientifically and technically.
Response: Thank you for your valuable comments and evaluation of our manuscript
However, some minor but important corrections are still required.
Specifically, in Figure 3, "kD" should be corrected to "kDa" to accurately reflect the standard unit of molecular weight.
Response: kD has been replaced by kDa in figure 3
Additionally, Figure 4 and Figure 5 lack scale bar measurements, which are essential for proper interpretation of immunofluorescence data. These issues must be corrected.
Response: The scale bar measurement has been indicated in the figure’s legend. The scale bar (white line represents 5.0 µm)
Reviewer 3 Report
Comments and Suggestions for Authors
See under the point Comments English quality
Comments on the Quality of English Language
The manuscript has been largely improved.
Some points however, should be tackled.
Line 658 "have showed"
Line 537 leave out "was"
Author Response
See under the point Comments English quality
Comments on the Quality of English Language
Response: The language has been greatly revised to meet the quality required by the journal.
The manuscript has been largely improved.
Response: Thank you for the comments
Some points however, should be tackled.
Line 658 "have showed"
Response: Addressed
Line 537 leave out "was"
Response: Addressed